# REVISIT FINETUNING STRATEGY FOR FEW-SHOT LEARNING TO TRANSFER THE EMDEDDINGS

## ABSTRACT

Few-Shot Learning (FSL) aims to learn a simple and effective bias on limited novel samples. Recently, many methods have been focused on re-training a randomly initialized linear classifier to adapt it to the novel features extracted by a pre-trained feature extractor (called Linear-Probing-based methods). These methods typically assumed the pre-trained feature extractor was robust enough, i.e., finetuning was not needed, and hence the pre-trained feature extractor does not be adapted to the novel samples. However, the unadapted pre-trained feature extractor distorts the features of novel samples because the robustness assumption may not hold, especially on the out-of-distribution samples. To extract the undistorted features, we designed Linear-Probing-Finetuning with Firth-Bias (LP-FT-FB) to yield an accurate bias on the limited samples for better finetuning the pre-trained feature extractor, providing stronger transferring ability. In LP-FT-FB, we further proposed inverse Firth Bias Reduction (i-FBR) to regularize the over-parameterized feature extractor on which FBR does not work well. The proposed i-FBR effectively alleviates the over-fitting problem of the feature extractor in the process of finetuning and helps extract undistorted novel features. To show the effectiveness of the designed LP-FT-FB, we conducted comprehensive experiments on the commonly used FSL datasets under different backbones for in-domain and cross-domain FSL tasks. The experimental results show that the proposed FT-LP-FB outperforms the SOTA FSL methods. The code is available at `https://github.com/whzyf951620/LinearProbingFinetuningFirthBias`.

## 1 INTRODUCTION

Few-shot Learning (FSL) has recently developed quickly in the limited data regime. FSL aims to learn a suitable inductive bias on the given limited samples of novel classes. At the very start, the whole model consisting of the feature extractor and the classifier is pre-trained on the samples of base classes, and then is finetuned on the limited novel samples to obtain an inductive bias. The performance of the finetuned model drops significantly due to the over-fitting problem of the pre-trained model. To address the over-fitting problem, meta-learning-based methods such as Prototypical Networks Snell et al. (2017) and MAML Finn et al. (2017) were proposed to learn the learning strategies for a suitable inductive bias. Then, Chen *et al.* proposed Baseline++ Chen et al. (2019) to show that a simple Linear Probing (LP) strategy can also get comparable performance to meta-learning-based methods. LP re-trained a linear classifier to adapt to the novel samples without updating the whole model. Following Baseline++, many researchers studied the LP-based FSL methods, such as S2M2 Mangla et al. (2020), RFS Tian et al. (2020), and EMD Zhang et al. (2020), are proposed to obtain a more powerful fully-trained feature extractor. LP-based FSL methods assumed that a pre-trained feature extractor is robust enough to novel samples Yang et al. (2021); Tian et al. (2020) and hence does not need to be finetuned.

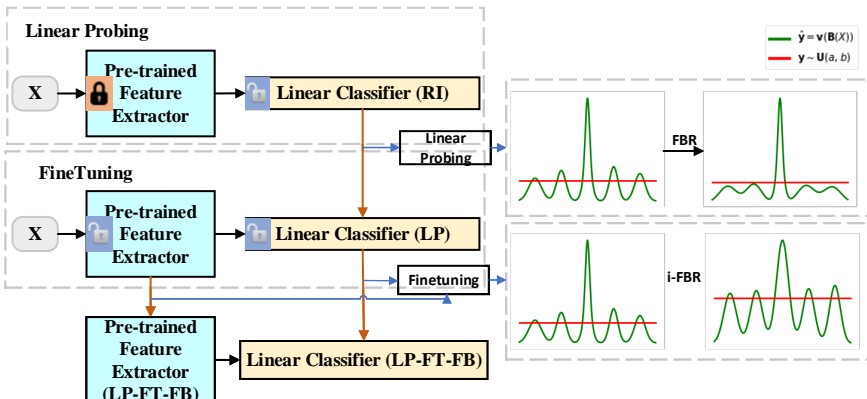

Figure 1: The proposed LP-FT-FB. The flow is divided into Linear Probing and FineTuning in two gray lines of dashes. (**Left**) At the linear probing stage, the parameters of the pre-trained feature extractor are frozen; at the finetuning stage, the parameters of the feature extractor are changeable. In the Linear Probing box, the **RI** means the linear classifier is randomly initialized. In the FineTuning box, the **LP** means the linear classifier is fully re-trained on the novel samples. (**Right**) The FBR is used to get an unbiased estimation for linear probing by encouraging the distribution of logits to be far away from the uniform distribution ($\mathbf{U}(a, b)$ in the figure). The proposed i-FBR addressed the over-fitting problem of the finetuned feature extractor by encouraging the distribution of logits to be close to the uniform distribution.

However, the robustness assumption may not hold, especially on the out-of-distribution (OOD) novel samples, because the existing regularization methods only provided effective robustness to in-distribution (ID) samples. This leads to "wrong" features of novel samples Kumar et al. (2022). This problem cannot be well addressed by only strengthening the robustness of the pre-trained feature extractor without transferring the extracted features. Finetuning is a typical and simple technology in the transferring learning on large-scale datasets to quickly transfer the features Kornblith et al. (2019); He et al. (2020); Chen et al. (2020). However, the existing FSL finetuning strategies cannot appropriately finetune the over-fitted feature extractor Shen et al. (2021). Consequently, *designing an appropriate **finetuning** strategy for the fully-trained feature extractor on the **limited samples*** is a critical point to transfer the features and hence make the finetuned feature extractor yield "good" novel features.

To design an appropriate finetuning strategy, we started by analyzing the existing finetuning strategies Kumar et al. (2022); Kanavati & Tsuneki (2021); Levine et al. (2016); Radford et al. (2021), especially on the ODD samples Kumar et al. (2022). Linear-Probing-Finetuning (LP-FT) Kumar et al. (2022) theoretically proved that the existing finetuning strategies underperformed due to distorted novel features. This is because when trying to fit the novel samples with a randomly initialized linear classifier, the extracted ID features and OOD features change inconsistently. The inconsistent problem is caused by the unchanged features of the samples in the space orthogonal to the space spanned by the base samples, i.e., out-of-distribution features unchanged (details in Section 2.2). Kumar *et al.* proposed using a linear classifier well-trained on the fitting novel samples to address this distorted problem.

Although LP-FT is effective, it is used to address the finetuning problem on large-scale out-of-distribution datasets, which contain enough samples to provide an unbiased estimation. In FSL tasks, limited novel samples lead to a biased estimation of the updated parameters. Seeing this, we reduce the Firth Bias of the model Ghaffari et al. (2022); FIRTH (1993) to obtain an unbiased estimation (called Firth Bias Reduction). However, Firth Bias Reduction (FBR) gives an unbiased estimation only for the linear classifier but not the feature extractor. Experimental result also shows that FBR decreased the performance of the finetuned feature extractor. A deep analysis (Section 2.3) shows that when the scaling factor $\lambda$ (see Eq. 2) is positive, FBR encourages the distribution of the linear classifier output to be far away from the uniform distribution (See Fig. 1) - FBR strengthens the influence of novel samples for linear probing. This accelerates the convergence

of the low-parameterized linear classifier but hampers the generalization of the over-parameterized feature extractor. According to the analysis, we proposed inverse-FBR (i-FRB, negative $\lambda$ in Eq. 2) to encourage the distribution of the linear classifier output to be close to uniform distribution (See Fig. 1) to address the over-fitting problem Müller et al. (2019).

Combining LP-FT, FBR, and the proposed i-FBR, we proposed Linear-Probing-Finetuning with Firth Bias (LP-FT-FB) to appropriately finetune the fully-trained feature extractors to adapt them to the target domain on limited novel samples. Firstly, we use linear probing to address the distorted feature problem. The linear probing is regularized with FBR for unbiased estimation. Then the whole pre-trained model is finetuned regularized with the proposed i-FBR on the limited novel samples to strengthen the transferring ability of the feature extractor for more separable features. The proposed LP-FT-FB makes the extracted features of novel samples undistorted and separable. The whole LP-FT-FB finetuning strategy is visualized in Fig. 1. Furthermore, the unbiased finetuned feature extractor transfers the novel features close to the novel domain. This is verified in the cross-domain FSL tasks (Section 3.5).

**Our main contributions** include (1) studying and finding the gaps between finetuning-based and LP-based FSL methods: distorted features and a biased estimation; (2) designing a novel and simple finetuning s-trategy, LP-FT-FB, to quickly transfer the extracted features to the novel domain to bridge the gaps; (3) proposing inverse Firth Bias Reduction (i-FBR) to address the biased estimation problem of finetuning the feature extractor due to limited samples.

## 2 METHODS

LP-FT is first described in Section 2.2. Then FBR is formulated and analyzed in Section 2.3 to address the unique limited-sample problem in FSL, i.e., the biased estimation problem. However, FBR is not suitable for the over-parameterized feature extractor. Based on the analysis of FBR, i-FBR is proposed to finetune the feature extractor. In Section 2.4, LP-FT-FB is stated.

### 2.1 SET UP

Following FBR Ghaffari et al. (2022), we assume a multinomial logitic regression model for the classifier $\mathbf{v}(\cdot, \boldsymbol{\beta})$. $\boldsymbol{\beta} = \{\beta_c\}_{1 \leq c \leq C}$ and $\beta_c$ denotes the logistic regression weights for class $c$. $\mathbf{B}(\cdot, \theta)$ denotes the feature extractor. The dataset $\mathcal{D} = \{(\mathbf{x}_i, \mathbf{y}_i)\}_{1 \leq i \leq M}$ totally contains $C$ classes and $M$ samples. The whole inference process is formulated as: $\hat{\mathbf{y}}_i = \mathbf{v}(\mathbf{z}_i, \boldsymbol{\beta}) = \mathbf{v}(\mathbf{B}(\mathbf{x}_i, \theta), \boldsymbol{\beta})$, where $\mathbf{z}_i = \mathbf{B}(\mathbf{x}_i, \theta)$. In the multinomial logistic regression model, the assignment probability of $\mathbf{z}_i$ to class $c$ is formulated as $p_i^c := Pr(\mathbf{y}_i = c | \mathbf{z}_i) = \frac{e^{\beta_c^T \mathbf{z}_i}}{1 + \sum_{c'=1}^{C} e^{\beta_{c'}^T \mathbf{z}_i}}$. Then the logistic log-likelihood function is formulated as $\mathcal{L}_{logistic} := \sum_{i=1}^{M} \sum_{c=1}^{C} \mathbf{1}[\mathbf{y}_i = c] \cdot \log p_i^c$, where $\mathbf{1}[\cdot]$ denotes the $\mathbf{y}_i$ in the one-hot manner. For the linear probing and finetuning, we used the Cross Entropy Loss $\mathcal{L}_{CE} = -\frac{1}{M} \sum_{i=1}^{M} [\mathbf{y}_i \cdot \log(\mathbf{P}_i)]$, where $\mathbf{P}_i = \{p_i^c\}_{c=1}^{C}$. $\mathbf{P}_i = \hat{\mathbf{y}}_i$ denotes the predicted probability vector.

### 2.2 LINEAR-PROBING-FINETUNING

LP-FT explored the reason why LP underperforms FT on in-distribution samples but outperforms FT in out-of-distribution samples on large-scale datasets. LP-FT theoretically proved that when a pre-trained feature extractor tried to fit the ID samples, the transfer of the model will not change the feature of out-of-distribution samples in the space orthogonal to the space spanned by pre-training samples.

To show this, let $\mathbf{S}$ denote the subspace spanned the training samples $\mathbf{X}$, and the training loss is $L(\mathbf{v}(\mathbf{B}(\mathbf{X}, \theta), \boldsymbol{\beta}), \mathbf{Y}) = \|\mathbf{X}\theta^T \boldsymbol{\beta} - \mathbf{Y}\|_2^2$. $\mathbf{B}$ is assumed to be a linear model; $\mathbf{Y}$ denotes the one-hot la-

bels of $\mathbf{X}$. The gradients of the training loss with respect to the parameter $\theta$ of the feature extractor $\mathbf{B}$ is computed as:

$$\nabla_\theta L(\mathbf{v}(\mathbf{B}(\mathbf{X}, \theta), \boldsymbol{\beta}), \mathbf{Y}) = 2\boldsymbol{\beta}(\mathbf{Y} - \mathbf{X}\theta^T\boldsymbol{\beta})^T\mathbf{X} \tag{1}$$

With Eq. 1, if $u$ is a sample in the subspace orthogonal to $\mathbf{S}$, the features of $u$ do not change with the finetuned $\mathbf{B}(\cdot, \theta)$: $\Delta\mathbf{z} = \nabla_\theta L(\mathbf{v}(\cdot, \boldsymbol{\beta}), \mathbf{B}(\cdot, \theta))u = 2\boldsymbol{\beta}(\mathbf{Y} - \mathbf{X}\theta^T\boldsymbol{\beta})^T(\mathbf{X} \cdot u) = 0$. This leads to the distorted features extracted by the finetuned $\mathbf{B}(\cdot, \theta)$ because the in-distribution and out-of-distribution features inconsistently change.

However, LP-FT explores the feature extractor finetuning problem on large-scale datasets. Observing this, we wish to address the feature extractor finetuning problem with LP-FT in FSL if we can address the limited-sample problem. Because the novel samples used to finetune the pre-trained feature extractor are out-of-distribution in FSL models. The limited-sample problem is actually a biased estimation problem.

### 2.3 INVERSE FIRTH BIAS REDUCTION

For linear probing, we use Firth Bias Reduction (FRB) to obtain an unbiased estimation. PMLE FIRTH (1993) added a log-determinant penalty [1] to remove the $\mathcal{O}(n^{-1})$ term of the asymptotic bias of the maximum likelihood estimate parameters for the unbiased estimation. FBR Ghaffari et al. (2022) addressed the situation when $\det(F) = 0$ and proposed an approximate format for the linear classifier and cosine classifier. In FBRGhaffari et al. (2022), the FBR loss is formulated as

$$\mathcal{L} = \mathcal{L}_{CE} + \mathcal{L}_{Firth} = \mathcal{L}_{CE} - \lambda \cdot \frac{1}{M}\sum_{i=1}^{M} D_{KL}(\mathbf{U}_{[0,C]}\|\mathbf{P}_i), \tag{2}$$

where $\mathbf{U}_{[0,C]}$ denotes the uniform distribution between 0 and $C$, and $\lambda$ denotes the scaling factor of FBR. From Eq. 2, we concluded that when $\lambda > 0$, the $\mathcal{L}_{Firth}$ encourages the distribution of the logits $\mathbf{z}_i$ to be far away from $\mathbf{U}_{[0,C]}$, which totally differs from Label Smoothing Müller et al. (2019). When $\lambda < 0$, $\mathcal{L}_{Firth}$ encourages the distribution of the logits $\mathbf{z}_i$ to be close to $\mathbf{U}_{[0,C]}$, and hence avoids too high confidence of $\mathbf{z}_i$. This coincides with the effectiveness of Label Smoothing Müller et al. (2019); Hein et al. (2019).

However, the vanilla FBR is not suitable for finetuning feature extractor because when $\lambda > 0$ (used in FBR Ghaffari et al. (2022)), the logits $\mathbf{z}$ is encouraged to be far away from $\mathbf{U}_{[0,C]}$. The distribution drag leads to too high similarity between the distribution of the logits $\mathbf{z}$ and $\mathbf{y}$, and hence makes the over-parameterized feature extractor over-fitted Hein et al. (2019). Seeing this, according to the analysis of Eq. 2, we proposed inverse FBR (i-FBR) to encourage the distribution of the $\mathbf{z}$ to be close to $\mathbf{U}_{[0,C]}$ with $\lambda < 0$ (we use the $\lambda_{inv}$ to denote the scaling factor of i-FBR) to address the over-fitting problem. The proposed i-FBR works similarly to Label Smoothing to a certain extent because both of them encourage the output distribution of the model to be close to the uniform distribution. Hence they can reduce the influence of the training novel samples and alleviate the too high confidence problem.

In addition to the above distribution drag, we computed the gradients of $\mathcal{L}_{CE}$ and $\mathcal{L}$ with respect to $\mathbf{r}_i = \boldsymbol{\beta} \cdot \mathbf{B}(\mathbf{x_i}, \theta)$ as follows to show the gradient variation brought by $\lambda$.

$$\frac{\partial\mathcal{L}}{\partial\mathbf{r}_i} = \frac{\partial\mathcal{L}_{CE}}{\partial\mathbf{r}_i} + \frac{\lambda}{(C+1)}(C\mathbf{P}_i - \mathbf{E}), \tag{3}$$

where $\mathbf{E}$ is a one-full vector with the same size as $\mathbf{P}_i$ and $\mathbf{P}_i = \hat{\mathbf{y}}_i$ is the predict of the whole model. The detailed derivation is given in Appendix. According to Eq. 3, we concluded that when $\lambda > 0$, the gradient $\frac{\partial\mathcal{L}}{\partial\mathbf{r}_i^c}$ is increased for the low-parameterized linear classifier to quickly reach the optimal point, and hence the linear probing model performs better Prabhu et al. (2021). The overfitting problem does not need to be considered because of the low-parameterized property.

---

[1]$\log(\det(F))$, where $F := -Hess_{\boldsymbol{\beta}}(\mathcal{L}_{logistic}) = \mathbb{E}_y[\boldsymbol{\nabla}_{\boldsymbol{\beta}}\mathcal{L}_{logistic} \cdot \boldsymbol{\nabla}_{\boldsymbol{\beta}}\mathcal{L}_{logistic}^T]$

However, it is not suitable for finetuning the over-parameterized feature extractor because of too large similarity between $z_i$ and $y_i$. From Eq. 3, we found that when $\lambda < 0$ and the class of $x_i$ is c, the gradient $\frac{\partial \mathcal{L}}{\partial \mathbf{r}_i^c}$ is decreased and $\frac{\partial \mathcal{L}}{\partial \mathbf{r}_i^k}$ $(k \neq c)$ is increased. This reduces the influence of $x_i$, which is used to train the over-parameterized feature extractor, and hence alleviates the over-fitting problem. Furthermore, different from Label Smoothing, the proposed i-FBR decreases the gradients, which is suitable for finetuning the over-parameterized model, and hence outperforms Label Smoothing (See Table 6).

## 2.4  LP-FT-FB

With FBR, LP-FT, and the proposed i-FBR, we proposed LP-FT-FB to quickly transfer the extracted features instead of the robustness assumption to address the over-fitting problem. Firstly, a randomly initialized linear classifier $\mathbf{v}(\cdot, \boldsymbol{\beta})$ is re-trained on the novel samples and regularized with FBR. For the $C_{few}$-*way*-K-*Shot* [2] FSL tasks, the re-trained classifier $\mathbf{v}'(\cdot, \boldsymbol{\beta}')$ is updated as

$$\boldsymbol{\beta}' = \boldsymbol{\beta} + \frac{\alpha_1}{K \cdot C_{few}} \cdot \sum_{i=1}^{K \cdot C_{few}} \frac{\partial \mathcal{L}(\hat{\mathbf{y}}_i, \mathbf{y}_i)}{\partial \mathbf{r}_i} \cdot \frac{\partial \mathbf{r}_i}{\partial \boldsymbol{\beta}}, \tag{4}$$

where $\mathbf{r}_i = \boldsymbol{\beta} \cdot \mathbf{B_0}(\mathbf{x}_i, \theta_0)$; $\hat{\mathbf{y}} = \{\hat{\mathbf{y}}_i\}_{i=1}^{C_{few} \cdot K}$; $\mathbf{y} = \{\mathbf{y}_i\}_{i=1}^{C_{few} \cdot K}$; $\alpha_1$ is the LP learning rate.

Then the pre-trained $\mathbf{B}_0(\cdot, \theta_0)$ and $\mathbf{v}'(\cdot, \boldsymbol{\beta}')$ are together finetuned on the same novel samples and regularized with i-FBR. FT is stated as follows. The forward process of FT is formulated as

$$\hat{\mathbf{y}}' = \mathbf{v}'(\mathbf{B}_0(\mathbf{x}, \theta_0), \boldsymbol{\beta}'). \tag{5}$$

The loss of FT is computed as:

$$\mathcal{L}(\hat{\mathbf{y}}', \mathbf{y}) = \frac{1}{C_{few} \cdot K} \sum_{i=1}^{K \cdot C_{few}} \mathcal{L}(\hat{y}_i', y_i). \tag{6}$$

The feature extractor $\mathbf{B}_0(\cdot, \theta_0)$ is updated as

$$\hat{\theta} = \theta_0 + \frac{\alpha_2}{K \cdot C_{few}} \cdot \sum_{i=1}^{K \cdot C_{few}} \frac{\partial \mathcal{L}(\hat{\mathbf{y}}_i', \mathbf{y}_i)}{\partial \mathbf{r}_i'} \cdot \frac{\partial \mathbf{r}_i'}{\partial \theta_0}, \tag{7}$$

where $\mathbf{r}' = \{\mathbf{r}_i'\}_{i=1}^{K \cdot C_{few}}$ and $\mathbf{r}_i' = \boldsymbol{\beta}' \cdot \mathbf{B_0}(\mathbf{x}_i, \theta_0)$. The linear classifier $\mathbf{v}'(\cdot, \boldsymbol{\beta}')$ is updated as

$$\hat{\boldsymbol{\beta}} = \boldsymbol{\beta}' + \frac{\alpha_2}{K \cdot C_{few}} \cdot \sum_{i=1}^{K \cdot C_{few}} \frac{\partial \mathcal{L}(\hat{\mathbf{y}}_i', \mathbf{y}_i)}{\partial \mathbf{r}_i'} \cdot \frac{\partial \mathbf{r}_i'}{\partial \boldsymbol{\beta}'}, \tag{8}$$

where $\alpha_2$ is the learning rate of FT. The analytic formula of Eq. 4, Eq. 8, and Eq. 7 are given in Appendix. With $\hat{\mathbf{B}}(\cdot, \hat{\theta})$ and $\hat{\mathbf{v}}(\cdot, \hat{\boldsymbol{\beta}})$, the performance of LP-FT-FB is evaluated on the novel query samples. The whole flow of LP-FT-FB is given in Appendix.

## 3  EXPERIMENTS

To comprehensively show the effectiveness of the proposed i-FBR and LP-FT-FB, we conducted many in-domain experiments: **1)** 5-way-1\5-shot tasks on *mini*-Imagenet, *tiered*-Imagenet, and CUB datasets under the typical FSL backbone, WideResNet-28-10; **2)** few-shot tasks under multiple scale backbones including ResNet-18 and ResNet-34; **3)** N-way-K-shot $(N > 5, K > 5)$ tasks on *mini*-Imagenet and *tiered*-Imagenet; **4)** few-shot tasks on the extracted features augmented by Distribution Calibration (DC) Yang et al. (2021). Also, the cross-domain few-shot tasks, *mini*-Imagenet $\rightarrow$ CUB and *tiered*-Imagenet $\rightarrow$ CUB, are also evaluated.

---

[2]Here, we used $C_{few}$-*way*-K-*Shot* to correspond with $C$ in Sec. 2.1, 2.2, and 2.3 instead of N-*way*-K-*Shot*.

### 3.1 EXPERIMENTAL SETUP

**1) Datasets.** The experiments are evaluated on three typical FSL datasets, *mini*-Imagenet Vinyals et al. (2016), *tiered*-Imagenet Ren et al. (2018), and CUB Wah et al. (2011). *mini*-Imagenet consists of 100 classes from the ImageNet, which are split randomly into 64 base, 16 validation, and 20 novel classes. Each class has 600 samples of size $84 \times 84$. *tiered*-Imagenet consists of 608 classes from the ImageNet, which are split randomly into 351 base, 97 validation, and 160 novel classes. CUB contains 200 classes with a total of 11,788 images of size $84 \times 84$. The base, validation, and novel split contain 100, 50, and 50 classes.
**2) Backbones.** To show the effectiveness of LP-FT-FB, we used many backbones. WideResNet-28-10 Zagoruyko & Komodakis (2016) with Dropout Hinton et al. (2012) is a typical backbone which used in the SOTA methods, such as S2M2 Mangla et al. (2020), FBR Ghaffari et al. (2022), and DC Yang et al. (2021). Also, multiple scale backbones including ResNet-18 He2 (2016) and ResNet-34, are used. The ResNets are the same as Mangla et al. (2020). **3) Finetuning hyper-parameters.** The pre-trained hyper-parameters are the same as Mangla et al. (2020). We only give the hyper-parameters in the LP-FT process. For LP, we used the linear classifier proposed in the Baseline++ Chen et al. (2019). For the optimizer, the SGD is used with the learning rate $\alpha_1 = 0.01$, momentum 0.9, dampening 0.9, and weight decay $1e^{-3}$. For the FBR of classifier, the factor $\lambda$ in Eq. 2 is set to 1. For FT, the feature extractor and the classifier are together manually finetuned with the learning rate $\alpha_2 = 1e^{-3}$ and the i-FBR factor $\lambda_{inv}$ in Eq. 7 is set to $-1e^{-3}$.
**4) Evaluation.** The reported results are averaged in percent with 95% confidence interval on 10000 tasks randomly selected by the Episodic Sampler. As the settings in Yang et al. (2021) and Ghaffari et al. (2022), the query sample number is 15 in each task for evaluating the performance.

### 3.2 5-WAY-1\5-SHOT TASKS

In-domain 5-way-1\5-shot tasks are the most typical FSL tasks. We directly used the pre-trained WideResNet-28-10 provided by $S2M2_R$ Mangla et al. (2020). The evaluated results are reported in Table 1. On *mini*-Imagenet, LP-FT-FB outperforms FBR by 1.45% and 0.86% for 1\5-shot tasks respectively. With the same pre-trained model, LP-FT-FB outperforms $S2M2_R$ by 2.56% and 1.23% for 1\5-shot tasks. Similarly, on the *tiered*-Imagenet, LP-FT-FB outperforms FBR by 3.04% and 3.75% for 1\5-shot tasks. On CUB, LP-FT-FB outperforms $S2M2_R$ by 1.48% and 1.03% for 1\5-shot tasks.

Table 1: The evaluation experiments for 5-*way* FSL tasks are conducted under the WideResNet28-10 on three typical FSL datasets.

| Methods | *mini*-Imagenet | | *tiered*-Imagenet | | CUB | |
|---|---|---|---|---|---|---|
| | 1-shot | 5-shot | 1-shot | 5-shot | 1-shot | 5-shot |
| MAML Finn et al. (2017) | 48.70 | 63.10 | – | – | 71.29 | 80.33 |
| ProtoNet Snell et al. (2017) | 54.16 | 73.68 | 65.65 | 83.40 | 71.88 | 87.42 |
| Baseline++ Chen et al. (2019) | 51.87 | 75.68 | 68.00 | 84.20 | 69.40 | 87.50 |
| S2M2 Mangla et al. (2020) | 64.48 | 82.67 | 73.50 | 88.00 | 80.68 | 90.85 |
| DeepEMD Zhang et al. (2020) | 65.91 | 82.41 | 71.16 | 86.03 | 75.65 | 88.69 |
| FBR Ghaffari et al. (2022) | 65.59 | 83.04 | 73.64 | 88.31 | 80.92 | 91.03 |
| LP-FT-FB | **67.04** | **83.90** | **76.68** | **92.06** | **82.16** | **91.88** |

### 3.3 EVALUATION UNDER DIFFERENT BACKBONES

In addition to WideResNet-28-10, we also evaluated LP-FT-FB under different backbones, ResNet-18 and ResNet-34. Because $S2M2_R$ did not provided the pre-trained models of ResNet-18 and ResNet-34, we used the code [3] of $S2M2_R$ to reproduce the pre-trained models.

---

[3] https://github.com/nupurkmr9/S2M2_fewshot

Table 2: The evaluation results for 5-*way* FSL tasks under different backbones.

| Dataset | Methods | ResNet-18 | | ResNet-34 | |
|---------|---------|-----------|-----------|-----------|-----------|
| | | 1-shot | 5-shot | 1-shot | 5-shot |
| *mini*-Imagenet | Baseline++ Chen et al. (2019) | 53.56 | 74.02 | 54.41 | 74.14 |
| | S2M2 Mangla et al. (2020) | 64.06 | 80.58 | 63.74 | 79.45 |
| | LP-FT-FB | **66.20** | **83.06** | **66.16** | **82.96** |
| CUB | Baseline++ Chen et al. (2019) | 67.68 | 82.26 | 68.09 | 83.16 |
| | S2M2 Mangla et al. (2020) | 71.43 | 85.55 | 72.92 | 86.55 |
| | LP-FT-FB | **73.36** | **86.88** | **76.48** | **90.02** |
| *tiered*-Imagenet | Baseline++ Chen et al. (2019) | 68.27 | 82.46 | 58.83 | 82.60 |
| | S2M2 Mangla et al. (2020) | 69.86 | 85.52 | 69.38 | 84.41 |
| | LP-FT-FB | **73.98** | **89.48** | **73.44** | **89.02** |

As given in Table 2, for ResNet-18, LP-FT-FB outperforms $S2M2_R$ by 2.14% and 2.48% for 1\5-shot tasks on *mini*-Imagenet. LP-FT-FB outperforms $S2M2_R$ by 1.93% and 1.33% for 1\5-shot tasks on CUB. For ResNet-34, LP-FT-FB outperforms $S2M2_R$ by 2.42% and 3.51% for 1\5-shot tasks on *mini*-Imagenet. LP-FT-FB outperforms $S2M2_R$ by 3.56% and 3.47% for 1\5-shot tasks on CUB.

## 3.4 EVALUATION ON AUGMENTED SAMPLES

The feature extractor is finetuned, and the novel features are since equivariant while novel samples totally differ from the base samples. The equivariant features are since closer to the distribution of the novel sample than the unchanged feature extractor used in DC. To show this, we used DC to augment the yielded features before evaluation. The evaluated results are given in Table 3.

The data augmented experiments are conducted on the *tiered*-Imagenet. For the 1-shot-10-way task, LP-FT-FB outperforms DC and FBR by 0.74% and 0.79%, respectively. For the 5-shot-10-way task, LP-FT-FB outperforms DC and FBR by 1.13% and 1.54%, respectively. Also, for the 15-way tasks, LP-FT-FB outperforms DC and FBR.

Table 3: The results of FSL tasks with DC data augmentation.

| Methods | *tiered*-Imagenet | | | |
|---------|------|------|------|------|
| | 1&10 | 5&10 | 1&15 | 5&15 |
| DC Yang et al. (2021) | 61.85 | 79.66 | 54.57 | 73.88 |
| FBR Ghaffari et al. (2022) | 61.90 | 80.07 | 54.62 | 74.40 |
| LP-FT-FB + DC | **62.64** | **81.20** | **55.02** | **75.64** |

## 3.5 CROSS-DOMAIN TASKS

To show the transferring ability brought by LP-FT-FB, we conducted cross-domain FSL experiments. LP-FT-FB is evaluated on the CUB dataset, but the feature extractor is pre-trained on *mini*-Imagenet and *tiered*-Imagenet, respectively. In this experiment, we used DC to augment the novel samples, just as the evaluated experiments in FBR Ghaffari et al. (2022). LP-FT-FB + DC denotes the proposed method in Table 4.

As is shown in Table 4, for *mini*-Imagenet → CUB tasks, LP-FT-FB outperforms DC by 2.07% and 2.29% for 5-shot-10-way and 5-shot-15-way tasks. For *tiered*-Imagenet → CUB tasks, LP-FT-FB outperforms DC by 1.65% and 2.09% for 5-shot-10-way and 5-shot-15-way tasks.

Table 4: The results of cross-domain FSL tasks. K & N denotes N-way-K-shot tasks.

| Methods | *mini*-Imagenet $\rightarrow$ CUB | | | | *tiered*-Imagenet $\rightarrow$ CUB | | | |
|---|---|---|---|---|---|---|---|---|
| | 1&10 | 5&10 | 1&15 | 5&15 | 1&10 | 5&10 | 1&15 | 5&15 |
| DC Yang et al. (2021) | 37.14 | 59.77 | 30.22 | 52.73 | 64.36 | 86.23 | 57.73 | 82.16 |
| FBR Ghaffari et al. (2022) | 37.40 | 60.77 | 30.37 | 53.84 | 64.52 | 86.66 | 57.74 | 83.05 |
| LP-FT-FB + DC | **37.96** | **61.84** | **30.98** | **55.02** | **65.02** | **87.88** | **58.26** | **84.25** |

## 3.6 ABLATION STUDY

**LP-FT *v.s.* FBR *v.s.* i-FBR.** We give the ablation study to explore the effectiveness of each part. The study is evaluated on *mini*-Imagenet and *tiered*-Imagenet for 5-way-1\5-shot tasks. As given in Table 5, we show the effectiveness of LP-FT-FB's each part.

Table 5: The ablation study on LP-FT *v.s.* FBR *v.s.* i-FBR.

| Strategy LP-FT | LP FBR | FT FBR | FT i-FBR | Datasets *mini*-Imagenet | *tiered*-Imagenet |
|---|---|---|---|---|---|
| ✓ | | | | 65.80 | 73.82 |
| | ✓ | | | 65.59 | 73.64 |
| ✓ | ✓ | | | 66.40 | 74.59 |
| ✓ | ✓ | | ✓ | **67.04** | **76.68** |
| ✓ | ✓ | ✓ | | 65.89 | 73.82 |
| ✓ | | | ✓ | 66.38 | 74.70 |

**i-FBR factor $\lambda_{inv}$ tuning.** We give the tuning of $\lambda_{inv}$ as follows to show the pattern of i-FBR.

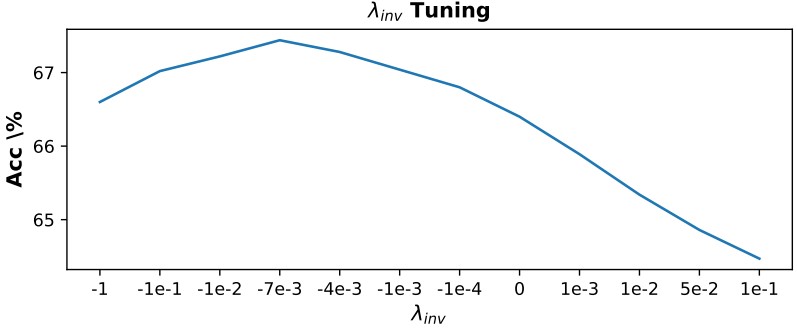

Figure 2: i-FBR factor $\lambda_{inv}$ tuning.

As given in Fig. 2, we show the tuning trend of $\lambda_{inv}$. The ablation experiments is evaluated on *mini*-Imagenet for the 5-way-1-shot task with $\lambda = 1$. We obtained the best performance with $\lambda_{inv} = -7e^{-4}$. While $\lambda_{inv} \geq 0$, i.e., i-FBR is turned into FBR, the performance is decreased.

**i-FBR *v.s.* smoothing regulariers.** The proposed i-FBR is a sort of regularizer smoothing the output distribution of the model. We compared the performance of different smoothing regularizers. As given in Table 6, the proposed i-FBR outperforms the smoothing regulaizers. The detailed experiment settings are given in Section Appendix.

Table 6: The ablation study on different smoothing regularizers. LP-FT-None denotes no smoothing regularizers is used.

| Methods | *mini*-Imagenet | | *tiered*-Imagenet | |
|---|---|---|---|---|
| | 1-shot | 5-shot | 1-shot | 5-shot |
| LP-FT-None | 66.40 | 83.20 | 75.66 | 90.85 |
| LP-FT-LS | 66.60 | 83.24 | 76.02 | 91.67 |
| LP-FT-Mixup | 66.64 | 83.37 | 76.00 | 91.54 |
| LP-FT-CutMix | 66.48 | 83.26 | 75.84 | 90.98 |
| LP-FT-FB | 67.04 | 83.90 | 76.68 | 92.06 |

## 4 RELATED WORKS

The FSL methods are divided into meta-learning-based methods and finetuning-based methods. The finetuning-based methods are divided into Linear-Probing-based and finetuning-the-whole methods.

**Finetuning-learning-based Methods.** In 2019, Chen *et al.* proposed Baseline++ Chen et al. (2019), which showed that a re-trained linear classifier could get a comparable performance to the meta-learning-based methods when the feature extractor is pre-trained. The pre-trained feature extractors are easy to be obtained without the need of training starting from scratch (used in meta-learning-based methods). Following Baseline++, Tian *et al.* proposed RFS Tian et al. (2020) to show that learning a supervised or self-supervised representation on the meta-training set, followed by training a linear classifier on top of this representation, gets a good performance better than most meta-learning-based methods. Similarly, S2M2 Mangla et al. (2020) and SKD Rajasegaran et al. (2020) proposed that the feature extractor pre-trained with Rotation Augmentation, Manifold Mixup Verma et al. (2019), and Knowledge Distillation Hinton et al. (2015) can make the feature extractor more robust. These methods are called Linear Probing methods because they only re-trained the linear classifier but fixed the pre-trained feature extractor, and they are based on a strong robustness assumption.

However, the robustness assumption may not hold. To address the problem, Shen *et al.* proposed Shen et al. (2021) a searching algorithm finding the layers suitable for finetuning. The searching algorithm is time-consuming and performs poorly on a single dataset because it does not address the distorted feature problem. Additionally, the related meta-learning FSL methods is left to Appendix.

## 5 CONCLUSION

In this paper, we proposed LP-FT-FB finetuning the pre-trained feature extractor to bring strong transferring ability to the feature extractor instead of a strong robustness assumption. The transferring ability addressed the distorted feature problem caused by the OOD novel samples. However, the strong transferring ability is at the cost of the over-parameterized feature extractor finetuning. The proposed LP-FT-FB is also time-consuming, just like the first-order gradient computation of Reptile Nichol et al. (2018). In future works, we will try to design a filter-wise or layer-wise finetuning method instead of the unit-wise one.

## 6 DECLARATIONS

This work was supported by the National Key R&D Program of China under Grant 2019YFF0302601, National Natural Science Foundation of China (No. 62071060), and the Beijing Key Laboratory of Work Safety and Intelligent Monitoring Foundation.

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
