# OpenReview forum: "Revisit Finetuning strategy for Few-Shot Learning to Transfer the Emdeddings"
_ICLR.cc/2023/Conference — ICLR 2023 poster_

### Official Review · Reviewer_rDwz · 2022-10-16

**Confidence:** 3
**Correctness:** 3
**Technical Novelty And Significance:** 3
**Empirical Novelty And Significance:** 3
**Recommendation:** 6

**Clarity, Quality, Novelty And Reproducibility:**

The paper reads smooth and the effectiveness of the method is "almost" well validated.

**Strength And Weaknesses:**

Pros:
1. The proposed method is well motivated and clearly stated.
2. The experimental results look good.


Cons:
1. The organization of the paper can be improved. (a) Table1 in the Supplementary Material is interesting. The reviewer suggests moving it to the main paper. Why not merge it with Table 1 in the Main paper?
2. The reviewer thinks adding experiments to show the results of different i-FBR factor lambda_inv might be interesting, ranging from values smaller than zero to values larger than zero.
3. Some experimental results are missing. Some experiments (Table.1) are conducted on mini-Imagenet, tiered-Imagenet, and CUB. But some only reported part of the results (e.g. in Table.2, the results of tiered-Imagenet are missing.). This makes the reader wondering whether the results of tiered-Imagenet do not "look perfect" and something is hidden behind. If possible, please add these kinds of missing resutls.
4. Minor issues: (1) The paper uses the format of ICLR 2021, instead of following the recent ICLR'2023 format. (2) The citations in the paper look strange, please fix them.


**Summary Of The Paper:**

This paper targets at few-shot classification. Motivated by Label Smooth, it designs LP-FT-FB to impose equivariance on the feature extractor to address the over-fitting problem. It first train a randomly initialized linear classifier on novel samples with FBR. Then the pretrained feature extractor and the classifier are finetuned on the same novel samples regularized with i-FBR. Experimental results show the effectiveness of the proposed method.

**Summary Of The Review:**

Although this paper has some minor weakness, considering its clear motivation, simplicity of the method, and good experimental results, the reviewer thinks the paper is above the acceptance threshold.

---

### Official Review · Reviewer_Yd7H · 2022-10-24

**Confidence:** 5
**Correctness:** 3
**Technical Novelty And Significance:** 3
**Empirical Novelty And Significance:** 3
**Recommendation:** 6

**Clarity, Quality, Novelty And Reproducibility:**

The clarity and novelty are somewhat satisfied. The quality of writing can be improved. The descriptions in the paper are clear for reproducibility.

**Strength And Weaknesses:**

Strengths:

    - It’s interesting to explore a better finetuning strategy for the few-shot learning problem.

    - The proposed method is clear to understand and easy to follow.

    - The method seems reasonable to obtain performance gain and the analysis and explanation are sufficient.

    - The improvement over other methods is satisfied.

Weaknesses:

    - Writing of this paper can be improved. Some sentences are colloquial and not scientific, for example: in the abstract, “we conducted a lot of experiments on the commonly …”. In related work “… and Knowledge Distillation Hinton et al. (2015) can make the feature extractor” is incomplete.

    - In Table 4, the authors provided the results using DC together with the proposed method, I wonder how about the results without DC. It's better to also provide this to demonstrate the effectiveness of the proposed method.

    - There is no ablation on each stage of the proposed learning procedure, for example:

     (1) train the linear classifier v(·, β) on the novel samples without FBR;

     (2) finetune B0(·, θ) and v0(·, β) together on the same novel samples without the proposed i-FBR. (better to put in the main paper instead of appendix)


**Summary Of The Paper:**

This paper discussed the distorted/biased features problem between base set and novel set in few-shot learning task, and proposed a Linear-Probing-Finetuning method with Firth-Bias to extract the undistorted features for shot-few learning. It also introduced an inverse Firth Bias Reduction (i-FBR) method for training the model in the few-shot learning setting. The learning procedures in the whole approach are as follows:

1)	pre-train backbone model B0(·, θ) on the base set with a regular loss function.

2)	a randomly initialized linear classifier v(·, β) is retrained on the novel samples and regularized with FBR.

3)	the pre-trained B0(·, θ) and v0(·, β) are together finetuned on the same novel samples and regularized with i-FBR.

Experiments in this paper are conducted on mini-Imagenet, tiered-Imagenet, and CUB datasets with ResNet-18, ResNet-34, and WideResNet28-10 backbones.


**Summary Of The Review:**

Overall, I think this is an interesting paper to explore the finetuning strategy for few-shot learning. However, I feel the writing can be improved with a few minor mistakes in the current shape, also some ablations on each component are encouraged to be put in the main text. I currently give borderline with the positive side. I'm open to increasing the rating according to the authors’ responses to my concerns.

---

### Official Review · Reviewer_C9jy · 2022-10-31

**Confidence:** 3
**Correctness:** 4
**Technical Novelty And Significance:** 2
**Empirical Novelty And Significance:** Not applicable
**Recommendation:** 6

**Clarity, Quality, Novelty And Reproducibility:**

Despite the aforementioned unclearness, overall the paper is easy to follow.

Please correct me if I am wrong about this. While the authors pointing out the issue of using FBR when fine-tuning the entire feature extractor is interesting, essentially, what the authors do was flip the coefficient of FBR from positive to negative, thereby decreasing the overfitting issue. For this, I would consider the paper somewhat novel but not entirely.

It seems the authors did not provide a link to their code for reproducibility.

**Strength And Weaknesses:**

Strengths
1. Provided an intuitive explanation alongside the mathematical proofs as to why the proposed inverse FBR works.
2. The authors compared the proposed method with many SoTA methods and showed consistent improvements over them.
3. One experiment demonstrated in Table 4 shows the method is complementary to one of the SoTA papers, Distribution Calibration.

Weaknesses
1. While the overall explanation seemed well-motivated, I had a question regarding the existence of samples that lie in the subspace orthogonal to the one spanned by the base samples. My concern is the following. If the feature extractor is pre-trained on the base classes, wouldn't the feature extractor tend to "put" whatever samples it sees in the subspace spanned by the pre-trained samples? I would appreciate more explanations from the authors pertaining to this argument.
2. What do the authors mean that in-distribution and out-of-distribution features inconsistently change? It's also not crystal clear to my mind although the authors do provide an explanation with a linear model.
3. The authors keep using "equivariance" in the article. However, for their scenario, e.g., using a feature trained on ImageNet on CUB200, it's not accurate (not sure if "invariance" would be a better choice). I would suggest the authors define what they mean by "equivariant" more clearly as what I would think of equivariance is when someone performs an operation on the image, the feature would change accordingly.

**Summary Of The Paper:**

This paper points out a potential issue with the linear-probing framework many SoTA methods are based on. Freezing a pre-trained feature encoder could get away with the feature inconsistency problem when one fine-tunes the feature extractor on the limited novel samples. However, the effectiveness of the frozen feature is built upon the assumption that the pre-trained feature extractor is robust enough for the novel samples, which nonetheless is usually not the case for out-of-distribution samples. The authors thus propose a method that effectively fine-tunes the features given limited samples from the novel classes. Specifically, on top of Firth bias reduction, they proposed an inverse FBR that decreases the influence of the novel classes when the extractor is fine-tuned.

**Summary Of The Review:**

Overall, this paper points out the issue of prevalent SoTA papers on few-shot learning and offers a solution, which the authors showed effective in the experiments compared with other SoTA methods. Despite the effectiveness of the proposed method, the method is slightly limited in terms of novelty as it is built upon the previous FBR paper. Summarizing all these perspectives, I would recommend a weak accept for now.

---

### Official Review · Reviewer_nALG · 2022-11-01

**Confidence:** 4
**Clarity, Quality, Novelty And Reproducibility:** 1) The latex template is from ICLR 20…
**Correctness:** 4
**Technical Novelty And Significance:** 2
**Empirical Novelty And Significance:** 2
**Recommendation:** 6

**Strength And Weaknesses:**


Strength:

1) This is the first time linear probing then fine-tuning is used for few-shot learning

2) The proposed method achieves state-of-the-art results among the fine-tuning methods (as opposed to meta-learning methods) on several few-shot learning benchmarks.


Weaknesses:

1) The results are outdated in comparison to the recent state-of-the-art in few-shot learning (For exemple: [3, 4, 5]). Only comparing with fine-tuning methods and not with all meta-learning baselines is unfair. Maybe an idea would be to use more recent pre-trained backbone, for exemple, how would it perform using similar backbones as in [6] ? Is there a reason to only use a pre-trained backbone from [7] ?

2) Other than inverting FBR, which is just using a negative coefficient for FBR, there is no conceptual novelty introduced in the paper. The linear probing then fine-tuning pipeline is introduced in [1] and FBR is introduced in [2].

3) A more in-depth ablation than what is provided in Appendix D is really necessary to understand the contribution of each component of the method. In particular, how does LP-FT performs without FRB in few-shot learning ?

4) The benchmarks used are standard, but very simple, both mini-imagenet and tired-imagenet use 5-way classification novel tasks, which is not challenging. It would be great to consider a much harder benchmark such as ImageNet-1k which has 311-way novel tasks. (See Table of [8], and Table 1.c) of [9]).


Questions and remarks:

1) How hard is it to tune $\lambda$ both in the linear probing step and in the fine-tuning step ? How do you do it in practice ? More generally, to add on the critic about the lack of ablation, it would be great to provide a full ablation on lambda on all configuration possible, i.e. $\lambda$ positive, negative and equal to 0, which correspond to FBR, i-FBR on no FBR, for both linear probing and fine-tuning steps (9 configurations in total).

2) Have you tried different probe than the linear probe ? For exemple non-parametric such as k-NN or 2-layer MLP ?

3) Would it be possible to use some kind of regularization on the weights of the linear probe, instead of FBR ? l2/l1-regularization, sparsity, ect.. ?

References

[1] A. Kumar et al., Fine-tuning can distort pretrained features and underperform out-of-distribution, ICLR 2022

[2] S. Ghaffari et al., On the importance of firth bias reduction in few-shot classification, ICLR 2022

[3] P. Rodriguez et al., Embedding Propagation: Smoother Manifold for Few-Shot Classification, ECCV 2020

[4] I. M. Ziko et al., Laplacian Regularized Few-Shot Learning, ICML 2020

[5] X. Chen et al., Few-Shot Learning by Integrating Spatial and Frequency Representation, CRV 2021

[6] S. Xu Hu et al., Pushing the Limits of Simple Pipelines for Few-Shot Learning: External Data and Fine-Tuning Make a Difference, CVPR 2022

[7] P. Mangla et al., Charting the right manifold: Manifold mixup for few-shot learning, CACV 2020

[8] Y-X. Wang et al., Low-Shot Learning from Imaginary Data, CVPR 2018

[9] L. Gui et al., Learning to Hallucinate Examples from Extrinsic and Intrinsic Supervision, ICCV 2021


**Summary Of The Paper:**

This paper proposes to use the "linear probing then fine-tuning" strategy (LP-FT) introduced in [1], as well as the firth bias reduction (FBR) introduced in [2], to learn the adaptation to novel-classes step in few-shot learning. The proposed method is evaluated on a variety of FSL tasks and demonstrate good results in the fine-tuning for FSL methods.

**Summary Of The Review:**

Combining LP-FT and FBR for few-shot learning is promising but the current results are far from competitive with recent approaches for few-shot learning. The paper does not introduce a new concept and therefore should at least demonstrate strong results. For these reasons I recommend a reject.

---

### Official Review · Reviewer_Qxg7 · 2022-11-03

**Confidence:** 3
**Clarity, Quality, Novelty And Reproducibility:** 1. The paper claims that  the propose…
**Correctness:** 3
**Technical Novelty And Significance:** 2
**Empirical Novelty And Significance:** 2
**Recommendation:** 6

**Strength And Weaknesses:**

1. The experiments, including those in the supplementary file, are extensive and show the effectiveness of the designed method.



**Summary Of The Paper:**

This paper aims to adapt LP-FT to Few shot learning and designs LP-FT-FB learning framework, in which inverse-FBR is proposed to regularize the feature extractor.  The experiments are extensive, but no theoretical proof is provided for the proposed i-FBR.



**Summary Of The Review:**

Without theoretical proof for i-FBR, I think the proposed method is not convincing and the novelty is limited.

---

### Decision · Program_Chairs · 2023-01-20

**Decision:**

Accept: poster

**Justification For Why Not Higher Score:**

All the reviewers feel the rating marginally above the acceptance threshold is fair for this paper, and a higher rating is too much. The main idea in this paper comes from the prior work on introducing Firth bias reduction to address the bias issue in few-shot classification. This paper extends the operation of Firth bias reduction from the pure classification layer to the feature backbone; and the proposed inverse Firth bias reduction simply changes the positive coefficient to be negative. There is also a lack of theoretical or rigorous analysis. The limitations make the area chairs cannot recommend a higher score.

**Justification For Why Not Lower Score:**

All reviewers recommend acceptance after discussion. There is no basis to overturn reviews. The area chairs agree with this recommendation. Given that the paper shows how to fine-tune the feature backbone appropriately without overfitting or biased estimation issues in few-shot learning, it should be of broad interest in the community.

In the camera ready, the authors should include the additional explanation and experiments presented in the discussion. Also, both the reviewers and area chairs feel that the use of “equivariance” in the paper is vague and not accurate, and it is suggested to remove this terminology in the camera ready. The authors should include the code for reproducibility purposes.


**Metareview: Summary, Strengths And Weaknesses:**

**Summary**:

The paper investigates how to fine-tune the feature backbone pre-trained on base classes to adapt to novel classes in the context of few-shot image classification. This is relevant to the biased estimation issue in small-sample regimes. The main idea is to extend the previous work of Firth bias reduction (which was proposed for unbiased estimation of classifier parameters) to inverse Firth bias reduction (flipping the coefficient from positive to negative) for the entire feature backbone fine-tuning. Specifically, a randomly-initialized linear classifier is first trained on novel samples regularized with Firth bias reduction. And then, the pre-trained feature extractor and the classifier are fine-tuned on the same novel samples regularized with inverse Firth bias reduction. Experiments on a variety of few-shot classification tasks show good results of the proposed method.

**Strengths And Weaknesses**:

The reviewers recognize the simplicity, effectiveness, and good results of this work.

The technical novelty of this work is a bit incremental, which leverages the existing work of Firth bias regularization. In addition, there is no strong theoretical justification. These issues remain after the revision.

Some ablations and experiments were missing in the original paper. The writing was not satisfactory, including the use of the ICLR 2021 template. The authors largely addressed these issues in the revision.

**Recommendation**:

All reviewers recommend acceptance after discussion. There is no basis to overturn reviews. The area chairs agree with this recommendation. In the camera ready, the authors should include the additional explanation and experiments presented in the discussion. Also, both the reviewers and area chairs feel that the use of “equivariance” in the paper is vague and not accurate, and it is suggested to remove this terminology in the camera ready. The authors should include the code for reproducibility purposes.


**Note From Pc:**

if the above contains the word "oral" or "spotlight" please see: "oral" presentation means -> notable-top-5% and "spotlight" means -> notable-top-25%. As stated in our emails, we are disassociating presentation type from AC recommendations

**Summary Of Ac-Reviewer Meeting:**

In the AC-reviewer meeting, Reviewer Yd7H pointed out that a common challenge in few-shot learning is how to fine-tune the backbone appropriately without overfitting. The paper demonstrates that it can achieve this objective, which should be of broad interest in the community. While Reviewer Yd7H was supportive, they also raised a question regarding if only novel examples are used for fine-tuning. The authors’ response later partially resolved this concern. So there is no remaining concern from Reviewer Yd7H.

Reviewer Qxg7 pointed out that the paper claims that the proposed inverse Firth bias reduction can address the biased estimation problem of ﬁne-tuning the feature extractor in few-shot learning. However, there is only experimental validation but no theoretical proof, which is not sufficiently convincing. The authors provided some clarification in the response, which helped to some extent. But the concern remains after the discussion. In addition, Reviewer Qxg7 would like to see the comparison with other smoothing regularization methods. The authors provided such a comparison and showed that the proposed approach is the best. Reviewer Qxg7 raised the score from marginally below the acceptance threshold to marginally above the acceptance threshold.

Reviewer C9jy initially gave the score marginally above the acceptance threshold, because the method is simple and intuitive. The main concern lies in that there is a lack of explanation why the method works – essentially, what is proposed is to flip the coefficient of Firth bias reduction from positive to negative, thereby decreasing the overfitting issue. For this, Reviewer C9jy considered the paper somewhat novel but not entirely. While the authors provided further explanation in the response, Reviewer C9jy was not fully satisfied. In addition, Reviewer C9jy felt that the use of equivariance here was not accurate.

Reviewer nALG’s main concerns are two-folds: (1) Other than inverting Firth bias reduction, which is just using a negative coefficient for Firth bias reduction, there is no conceptual novelty introduced in the paper. Both Firth bias reduction and the linear probing then fine-tuning pipeline have been introduced before. (2) Large-scale experiments and some ablations are missing. In the response, the authors successfully addressed the evaluation concern, and clarified the difference between transductive and inductive few-shot settings. Reviewer nALG raised the score to be marginally above the acceptance threshold.

At the end of the meeting, all reviewers reached a consensus of scores: marginally above the acceptance threshold. The area chairs agree with this recommendation.